# Robustifying Generalizable Implicit Shape Networks with a Tunable Non-Parametric Model

**Amine Ouasfi**     **Adnane Boukhayma**
Inria, Univ. Rennes, CNRS, IRISA, M2S, France

## Abstract

Feedforward generalizable models for implicit shape reconstruction from unoriented point cloud present multiple advantages, including high performance and inference speed. However, they still suffer from generalization issues, ranging from underfitting the input point cloud, to misrepresenting samples outside of the training data distribution, or with toplogies unseen at training. We propose here an efficient mechanism to remedy some of these limitations at test time. We combine the inter-shape data prior of the network with an intra-shape regularization prior of a Nyström Kernel Ridge Regression, that we further adapt by fitting its hyperprameters to the current shape. The resulting shape function defined in a shape specific Reproducing Kernel Hilbert Space benefits from desirable stability and efficiency properties and grants a shape adaptive expressiveness-robustness trade-off. We demonstrate the improvement obtained through our method with respect to baselines and the state-of-the-art using synthetic and real data.

## 1 Introduction

Enabling machines to understand and navigate 3D is a major challenge. Instances of this ability include downstream tasks in computer vision and graphics, such as shape reconstruction from noisy, incomplete and relatively sparse point clouds. Building full shapes from point clouds is all the more an important problem in account of the ubiquity of this light, albeit incomplete, 3D shape representation, whether it is acquired from the increasingly democratized consumer grade and industrial depth sensors or *e.g.* as obtained from photogrammetry (*e.g.* Structure From Motion, Multi-View Stereo [62, 63]). Whilst Classical optimization based approaches such as Poisson Reconstruction [33] and Moving least squares [27] can deliver mostly good reconstructions for all intents and purposes, they require still dense clean point sets with reliable normal estimations. More recently, deep learning based alternatives have been shown to offer faster and more robust predictions especially for noisy and sparse inputs, without requiring normal information.

Within this class of methods, state-of-the-art feed forward (test time optimization-free) generalizable models are based on implicit neural shape representations, where explicit meshes can be extracted at test time using Marching Cubes [43]. While they offer many desirable properties, such as all-round good performance and fast inference (within tens of milliseconds on a standard GPU), they still suffer from limitations, some of which we come across upon examining the state-of-the-art method [7] for instance:

**Limited OOD generalization** As these models are typically trained with full 3D supervision, training commonly uses synthetic data such as the ShapeNet [12] dataset, large real 3D supervision being famously hard to come by. This results in a performance drop for real test data outside the training distribution. Comparing Tables 2 and 6, the performance of [7] drops by roughly fourfold going from testing on synthetic data to real scans.

**Limited input size generalization** As illustrated in table 7, when presented with testing point clouds that are larger (*e.g.* 10k) than the training size (3k), the performance does not scale and even deteriorates compared to testing on the training size.

37th Conference on Neural Information Processing Systems (NeurIPS 2023).

**Underfitting the input** There is no explicit guaranty the input points are at the predicted shape function level set from a single forward pass. We hypothesise that enforcing this constraint under well chosen regularizations could potentially lead to improvements at test time and a reduction of the generalization issues mentioned above.

We want to formulate an efficient strategy to improve on these aspects of generalizable shape networks at test time, *i.e.* reconstructing shapes by dynamically adjusting the network to the input point-cloud during testing. We note that we follow such a strategy as training large models from scratch can be resource intensive, while adaptation can be less costly and has been shown to yield considerable improvements in *e.g.* domain adaptation and transfer learning literature. One possible solution [69] (SAC) is to finetune the network weights at test time, using the knowledge that input point cloud samples are on the desired surface as supervision. However, we find this strategy to be unstable as it can overfit on these samples. As shown throughout the results (Section 4), it even exacerbates the performance of its initial baseline network in many cases. This instability highlights the necessity for regularization in this adaptation process. We need to balance the need to maintain the expressive capacity of the learned functions while ensuring stability and robustness. To this end we devise three important design choices:

• We restrain the hypothesis space of the adapted shape functions to be a Reproducing Kernel Hilbert Space (RKHS). By the Representer Theorem, the minimizer of our regularized empirical risk minimization (ERM) problem (Equation 7) emerges naturally as the solution of a Kernel Ridge Regression (KRR) problem.

• By using a Gaussian kernel in this context, we benefit from the universality properties of the associated (RKHS) *i.e.* a hypothesis space rich enough to approximate any continuous function arbitrarily well. Thus, we avoid the difficulties arising from optimizing a large number of neural network parameters, as in SAC, while maintaining the expressive capacity necessary for effective shape modeling.

• We propose a strategy to solve our regularized ERM problem in a RHKS space adapted to the shape we would like to recover. This is possible thanks to the unique correspondence between RKHS and kernels. Hence, instead of relying on handmade fixed kernels, we learn the KRR hyperparameters using a loss function that avoids overfitting on the data term (Equation 11).

In this regard, approximate KRR solvers based on Nyström samples (Nyström KRR) [48] can be computed efficiently and allow scaling in point cloud size. As shown in our ablation studies, defining this kernel in euclidean space seems insufficient to constrain the fitting problem. Hence we define it in the feature space of the pretrained convolutional shape network. To define the fitting data, we use the input point cloud with their inherent *label*: being samples from the surface by default. We also augment the fitting data with additional samples paired with their shape network predictions as pseudo-labels. This augmentation ensures the fitting problem is well defined. We note that through this strategy, knowledge is being transferred from the network to our kernel regression through both features and pseudo-labels.

Until recently, exiting methods for KRR hyperparameter tuning, *e.g.* cross-validation or grid search, placed strong limits on the flexibility and number of hyperparameters that can be tuned. The work of Meanti *et al.* [47] proposed lately a automated gradient based tuning of the hyperparameters of the KRR by minimizing a loss that prevents overfitting on the fitting data. This strategy allows the optimization of both the kernel hyperprameters and the Nyström samples, and proves beneficial in our context, *i.e.* fitting a shape function in feature space, as compared to baselines with fixed Nyström samples and/or kernel hyperparameters (See ablation in section 4.7). A visual summary of our method can be found in Figure 1.

To test our idea we devise experiments on real and synthetic data targeting generalization issues. Our approach improves consistently on the baseline networks, and outperforms other state-of-the-art methods and test time network finetuning based strategies. We also show that our approach can be applied successfully to more than one network. Our ablation studies showcase also the importance of all the various components of our method.

## 2 Related Work

We review in this section previous work we deemed most relevant to our problem and contribution.

**Shape Representations in Deep Learning**    Shapes can be represented within deep learning either intrinsically or extrinsically. Intrinsic representations are a discrimination of the shape itself. When done explicitly, using *e.g.* tetrahedral or polygonal meshes [73, 32] or point clouds [22], the output topology in predefined thus bounding the variability of the shapes that can be generated. Among other forms of intrinsic representations, 2D patches [26, 79, 20] can prompt discontinuities, whilst the simplicity of shape primitives such as cuboids [72, 86], planes [39] and Gaussians [24] limits their expressiveness. Differently, extrinsic shape representations model the entire space containing the scene or object of interest. Voxel grids [82, 81] are the most popular one being the direct extension of 2D pixels to 3D domain. However, their capacity is limited by their cubic resolution memory cost. Sparse representations such as octrees [60, 70, 75] can alleviate this issue to some extent.

**Implicit Neural Shape Representations**    Implicit neural representations (INRs) emerged recently as a major medium for modelling extrinsic shape and radiance fields (*e.g.* [51, 83, 74, 30, 11]). They overcome many of the limitations of the aforementioned classical representations thanks to their ability to represent shapes with arbitrary topologies at virtually infinite resolution. They are usually parameterised with MLPs mapping spatial locations or features to *e.g.* occupancy [50], signed [55] or unsigned [16, 85] distances relative to the target shape. The level-set of the inferred field from these MLPs can be rendered through ray marching [28], or tessellated into an explicit shape using *e.g.* Marching Cubes [43]. Another noteworthy branch of work builds hybrid implicit/explicit representations [54, 14, 19, 84] based mostly on differentiable space partitioning. In order to represent collections of shape simultaneously, implicit neural models require conditioning mechanisms. These include feature and latent code concatenation, batch normalization, hypernetworks [65, 67, 66, 76, 13] and gradient-based meta-learning [52, 64]. Concatenation based conditioning was first implemented using single global latent codes [50, 15, 55], and further improved with the use of local features [53, 35, 23, 71, 68, 57, 17, 31, 21].

**Reconstruction from Point Cloud**    Classical approaches include combinatorical ones where the shape is defined through an input point cloud based space partitioning, through *e.g.* alpha shapes [5] Voronoi diagrams [1] or triangulation [9, 41, 59]. On the other hand, the input samples can be used to define an implicit function whose zero level set represents the target shape, using global smoothing priors [78, 36, 80] *e.g.* radial basis function [8] and Gaussian kernel fitting [61], local smoothing priors such as moving least squares [49, 27, 34, 42], or by solving a boundary conditioned Poisson equation [33]. The recent literature proposes to parameterise these implicit functions with deep neural networks and learn their parameters with gradient descent, either in a supervised or unsupervised manner.

**Unsupervised Implicit Neural Reconstruction**    A neural network is typically fitted to the input point cloud without extra information. Regularizations can improve the convergence such as the spatial gradient constraint based on the Eikonal equation introduced by Gropp *et al.* [25], a spatial divergence constraint as in [4], Lipschitz regularization on the network [40]. Atzmon *et al.* learns an SDF from unsigned distances [2], and further supervises the spatial gradient of the function with normals [3]. Ma *et al.* [44] expresses the nearest point on the surface as a function of the neural signed distance and its gradient. They also leverage self-supervised local priors to deal with very sparse inputs [45] and improve generalization [46]. All of the aforementioned work benefits from efficient gradient computation through back-propagation in the neural network. Periodic activations were introduced in [65]. Lipman [38] learns a function that converges to occupancy while its log transform converges to a distance function. [80] learns infinitely wide shallow MLPs as random feature kernels using points and their normals. Most of the aforementioned methods present failures under sparse and noisy input due to the lack of supervision and data priors. Differently, we propose here to combine a data prior based method and self-supervised learning.

**Supervised Implicit Neural Reconstruction**    Supervised methods assume a labeled training data corpus commonly in the form of dense samples with ground truth shape information. Auto-decoding methods [35, 55, 71, 31, 10] require test time optimization to be fitted to a new point cloud, which can take up to several seconds. Encoder-decoder based methods enable fast feed forward inference. Introduced first in this respect, Pooling-based set encoders [50, 15, 23] such as PointNet [58] have been shown to underfit the context. Convolutional encoders yield state-of-the-art performances. They use local features either defined in explicit volumes and planes [57, 17, 37, 56] or solely at the input points [7]. Ouasfi *et al.* [53] proposed concurrently the first convolution-free fast feed forward generalizable model. Peng *et al.* [56] proposed a differentiable Poisson solving layer that converts predicted normals into an indicator function grid efficiently. However, it is limited to small scenes

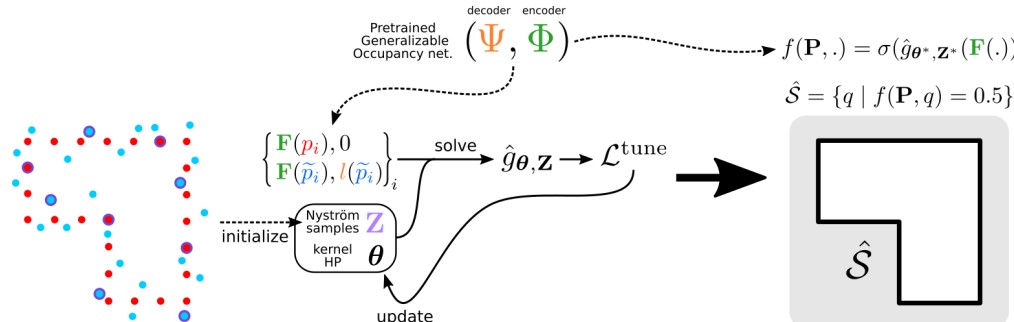

Figure 1: Overview. Our method predicts an implicit shape function from a noisy unoriented input point cloud. We combine a cross-shape deep prior (Pretrained generalizable occupancy network) and an intra-shape adaptive Nyström Kernel Ridge Regression (NKRR) $g$ at test time. The latter learns to map network features of the input points and additional points to the level-set and network generated pseudo-labels respectively. The NKRR hyperparameters ($\mathbf{Z}$,$\boldsymbol{\theta}$) are adjusted to the current shape.

due to the cubic memory requirement in grid resolution. Most of these methods still suffer from generalization issues. Concurrently, the work in [78, 29] proposes to build generalizable models where the implicit decoder is a kernel regression. Differently, we advocate a strategy to improve the generalization of any preexisting pretrained reconstruction network.

# 3 Method

Our input is a noisy unoriented point cloud $\mathbf{P} \subset \mathbb{R}^{3 \times N_P}$. Our objective is to recover a 3D reconstruction, *i.e.* the shape surface $\mathcal{S}$ that best explains this observation, where the input point cloud elements approximate noisy samples from $\mathcal{S}$.

To achieve this, we model an implicit function $f$ that predicts occupancy values relative to a target shape $\mathcal{S}$ at any queried euclidean space location $q \in \mathbb{R}^3$, given the input point cloud $\mathbf{P}$, *i.e.* $f(\mathbf{P}, q) = 1$ if $q$ is inside the shape, and $0$ otherwise. The inferred shape $\hat{\mathcal{S}}$ can then be obtained as a level set of the occupancy field inferred through $f$:

$$\hat{\mathcal{S}} = \{q \in \mathbb{R}^3 \mid f(\mathbf{P}, q) = 0.5\}. \tag{1}$$

In practice, an explicit triangle mesh for $\hat{\mathcal{S}}$ is extracted using the Marching Cubes [43] algorithm.

## 3.1 Feedforward Generalizable Model

The staple back-bone models for feedforward generalizable implicit shape reconstruction from point cloud (*e.g.* [57, 17, 7]) have typically a two-stage architecture. First, a deep convolutional network (usually a U-Net) builds features in $\mathbb{R}^{N_F}$ from the input point cloud:

$$\mathbf{F} = \Phi(\mathbf{P}). \tag{2}$$

Features $\mathbf{F}$ can be extrinsic or intrinsic. For instance in [57, 17], the ConvNet builds an extrinsic 2D or 3D explicit feature grid representing space around the shape. More recently, Boulch *et al.* [7] argued against this strategy, contending that voxel centers may be far from the input point cloud locations, and are sampled uniformly while they should be more focused near the surface of interest. They hence define features intrinsically, *i.e.* only at the input point cloud locations. In both strategies, the ConvNet offers transnational equivariance and generalization ability. It is worth noting that it is also naturally endowed with a locality mechanism, which is crucial for performance with fast inference times, as only a single forward pass in the encoder is needed to build the feature space. Conversely, other generalizable models lacking locality in their encoders, such as the work in [21], enforce locality through local patch inputs. They hence require as many encoder forward passes as there are query points. This naturally results in significantly increased inference times compared the the aforementioned competition that we build on.

Given a 3D query point $q$, an implicit decoder, typically an MLP, maps the feature to the occupancy value:

$$\text{occ}(q) = \sigma(\Psi(\mathbf{F}(q))) \tag{3}$$
$$= \sigma(l(q)), \tag{4}$$

where $\sigma(.)$ represents the Sigmoid activation and $l(.)$ a logit function. In the case of extrinsic feature networks [57, 17, 56], feature $\mathbf{F}(q)$ is obtained by linear interpolation (trilinear in the case of 3D feature grids, and bilinear for 2D grids). For intrinsic feature networks [7], the feature is pooled order-invariantly from the nearest points of the input point cloud. Without loss of generality we build in this work on the model in [7], as it has been shown to achieve state-of-the-art performances on several benchmarks.

## 3.2 Kernel Ridge Regression as a Shape Function in Feature Space

Existing pretrained generalizable feedforward models for reconstruction from point cloud suffer from generalization limitations. As they are typically trained on synthetic shape datasets (*e.g.* ShapeNet [12]) with full supervision, using a predefined point cloud size and simulated noise, their performance reduces when tested on point clouds that are different from the training ones regarding properties such as density, and whether they are real or simulated. Such shortcomings of an instance of these models are showcased in Tables 7 and 5. We design henceforth an efficient mechanism that can fill this performance gap, while benefiting from the available shape labels at test time: The fact that the input point cloud is made of samples that belong to the surface we wish to reconstruct.

We consider the following training data samples:

$$\{(x_i, y_i)\}_{i=1}^n = \{(\mathbf{F}(p_i), 0)\}_{i=1}^{N_p} \cup \{(\mathbf{F}(\widetilde{p}_i), l(\widetilde{p}_i))\}_{i=1}^{N_p}, \tag{5}$$

where $(x_i, y_i) \in \mathbb{R}^{N_F} \times \mathbb{R}$ and $n = 2 \times N_p$. $\{p_i\}$ are the samples of the input point cloud $\mathbf{P}$. $\{\widetilde{p}_i\}$ are additional samples around this point cloud, that can be automatically generated following *e.g.* the strategy in [44]. Features $\mathbf{F}(.)$ and occupancy logits $l(.)$ are obtained from the pretrained convolutional occupancy model (See Equations 2 and 4). We note that these data samples are corrupted by a combination of observation noise in the input point cloud (points $p_i$ not being exactly at the surface) and prediction errors of the initial model (pseudo-labels $l(\widetilde{p}_i)$ being inaccurate).

We approximate the mapping from deep shape features $x_i$ to occupancy logits $y_i$ using a non-linear non-parametric regression $g(.)$ to provide spatial regularization while favouring flexibility, *i.e.* without making strong assumptions on the model. In particular, we use Kernel Ridge Regression (KRR), where the hypothesis space of the mapping is a reproducing kernel Hilbert space $\mathcal{H}$. Associated to this space is a kernel function $k_{\boldsymbol{\theta}} : \mathbb{R}^{N_F} \times \mathbb{R}^{N_F} \to \mathbb{R}$, that depends on hyperparameters $\boldsymbol{\theta}$. To ensure the minimisation of the square error of the mapping over the training samples is well defined, a regularization is typically needed, with a weight $\lambda$:

$$\hat{g}_{\boldsymbol{\theta}} = \underset{g \in \mathcal{H}}{\arg\min} \|g(\mathbf{X}) - \mathbf{Y}\|^2 + \lambda \|g\|_{\mathcal{H}}^2, \tag{6}$$
$$\text{where} \quad \mathbf{X} = \{x_i\}_{i=1}^n, \quad \mathbf{Y} = \{y_i\}_{i=1}^n. \tag{7}$$

The unique solution to this problem is expensive to compute. For efficiency we follow the approximation to KRR that considers a lower subspace of dimension $m << n$ [77]. This allows our method to scale to large input point clouds. The approximation uses $m$ inducing features $\mathbf{Z} = \{z_j\}_{j=1}^m$, also known as Nyström Samples, where $z_j \in \mathbb{R}^{N_F}$.

Finally, the unique solution to the minimization in Equation 6 writes [48]:

$$\hat{g}_{\boldsymbol{\theta}, \mathbf{Z}} = \sum_{j=1}^m \beta_j k_{\boldsymbol{\theta}}(\cdot, z_j), \tag{8}$$

$$\text{where} \quad \boldsymbol{\beta} = \left(\mathbf{K}_{nm}^\top \mathbf{K}_{nm} + \lambda n \mathbf{K}_{mm}\right)^{-1} \mathbf{K}_{nm}^\top \mathbf{Y}. \tag{9}$$

Here, $(\mathbf{K}_{nm})_{i,j} = k_{\boldsymbol{\theta}}(x_i, z_j)$ and $(\mathbf{K}_{mm})_{i,j} = k_{\boldsymbol{\theta}}(z_i, z_j)$, where $i \in [\![1, n]\!]$ and $j \in [\![1, m]\!]$. This solution can be computed efficiently through *e.g.* the Falcon solver [48].

### 3.3 Learning the Shape Function Hyperparameters

Next, based on [47], we propose to automatically learn the hyperparameters of our Nyström KRR, namely the kernel hyperparameters $\boldsymbol{\theta}$ and the Nyström samples $\mathbf{Z}$, using gradient descent. This shape specific tuning allows for a larger hyperparameter space and using more expressive kernels. The optimization consists in minimizing an upper bound $\mathcal{L}^{\text{tune}}$ of the ideal objective, *i.e.* the expectation over all possible data points of the squared error of the model. This allows to avoid overfitting on the training set:

$$\mathcal{L}^{\text{tune}} = \frac{2}{n}\operatorname{Tr}\left((\widetilde{\mathbf{K}} + n\lambda\mathbf{K})^{-1}\widetilde{\mathbf{K}}\right) + \frac{2}{n\lambda}\operatorname{Tr}(\mathbf{K} - \widetilde{\mathbf{K}})\mathcal{L}^{\text{data}}(\hat{g}_{\boldsymbol{\theta},\mathbf{z}}) + 2\mathcal{L}^{\text{data}}(\hat{g}_{\boldsymbol{\theta},\mathbf{z}}), \quad (10)$$

$$\text{where} \quad \mathcal{L}^{\text{data}}(\hat{g}_{\boldsymbol{\theta},\mathbf{z}}) = \frac{1}{n}\|\hat{g}_{\boldsymbol{\theta},\mathbf{z}}(\mathbf{X}) - \mathbf{Y}\|^2 + \lambda\|\hat{g}_{\boldsymbol{\theta},\mathbf{z}}\|_{\mathcal{H}}^2. \quad (11)$$

$\widetilde{\mathbf{K}} = \mathbf{K}_{nm}\mathbf{K}_{mm}^{\dagger}\mathbf{K}_{nm}^{\top}$ and $\dagger$ symbolizes the Moore-Penrose matrix inverse. The upper bound is made, in this order, of he Nyström penalty attempting to make the Nyström kernel $\widetilde{\mathbf{K}}$ closer to the complete kernel $\mathbf{K}$, a second term penalizing overly complex kernels, and finally a data fit term. We note that the authors in [47] derived this loss by replacing expectations with their empirical counterparts. They also propose to use the Hutchinson approximation to compute matrix traces for efficiency.

Aftre $N$ gradient descent steps (as illustrated in Algorithm 1), we select hyperparameters $\boldsymbol{\theta}^*$ and $\mathbf{Z}^*$ that minimize the squared error of the model over the training data following [47]. For a given query point q, our final implicit occupancy function can be expressed as the sigmoid activated optimal Nyström KRR in feature space:

$$f(\mathbf{P}, q) = \sigma(\hat{g}_{\boldsymbol{\theta}^*, \mathbf{z}^*}(\mathbf{F}(q))) \quad (12)$$

---

**Algorithm 1** The training procedure of our method.

---

**Input:** Point cloud $\mathbf{P}$, pretrained occupancy network $(\Psi, \Phi)$, learning rate $\alpha$
**Output:** optimal kernel hyperparameters $\boldsymbol{\theta}^*$ and Nyström samples $\mathbf{Z}^*$

$\quad \widetilde{\mathbf{P}} = \text{upsample}(\mathbf{P})$
$\quad \mathbf{X} = \Phi([\mathbf{P}, \widetilde{\mathbf{P}}])$
$\quad \mathbf{Y} = [\mathbf{0}, \Psi \circ \Phi(\widetilde{\mathbf{P}})]$
$\quad$ initialize $\mathbf{Z}$ as $m$ random features from $\mathbf{X}$, initialize $\boldsymbol{\theta}$
$\quad$ **for** $N$ times **do**
$\quad\quad$ compute $\hat{g}_{\boldsymbol{\theta},\mathbf{z}}$ from $\mathbf{X}$, $\mathbf{Y}$ (Equations 8 and 9)
$\quad\quad$ compute $\mathcal{L}^{\text{tune}}$ for $\hat{g}_{\boldsymbol{\theta},\mathbf{z}}$, $\mathbf{X}$ and $\mathbf{Y}$ (Equations 10 and 11)
$\quad\quad (\boldsymbol{\theta}, \mathbf{Z}) \leftarrow (\boldsymbol{\theta}, \mathbf{Z}) - \alpha\nabla_{\boldsymbol{\theta},\mathbf{z}}\mathcal{L}^{\text{tune}}$
$\quad$ **end for**

---

## 4 Results

In this section, we evaluate our method using real and synthetic data both quantitatively and qualitatively. As our method builds on Poco [7], we compare our results to this baseline. We also compare to other state-of-the-art convolutional occupancy networks, such as Conv [57] and SAP [56]. We compare to the strategy based on test-time finetuning of a convolutional occupancy network (Conv [57] ) in SAC [69]. Additionally, we compare to the strategy based on fitting an MLP to the point cloud in NP [44]. We compare also to classical learning-free Poisson reconstruction (SPSR [33]) where input point normals are estimated with PCA. For state-of-the-art convolutional occupancy networks Poco, Conv and SAP, we use their official publicly available pretrained models as trained on the same training split of ShapeNet [12] with 3k sized point clouds and noise of variance 0.005. We use the publicly available official implementations of SAP and NP. We present also an ablative analysis of the components of our method in Section 4.7. We provide additional results in the supplementary material. We note that from a computational stand point, for a 10k sized input point cloud, our method takes roughly 10 seconds to converge on a NVIDIA RTX A6000 GPU. The other test-time tuning based alternatives we compare to here, namely SAC and NP take roughly 1 and 5 minutes respectively.

### 4.1 Implementation Details

We run all the experiments for 100 epochs using Adam optimizer with learning rate 0.1. We use the Falcon Library to implement the tuning of Nyström KRR in feature space. Regarding the input size and feature dimensions, We experimented with $N_P = 10k$ (also 500 and 3k in ablation) and $N_F = 32$ and $m = 500$ inducing points.

### 4.2 Metrics

Following seminal work, we evaluate our method and the competition w.r.t. the ground truth using standard metrics for the 3D reconstruction task. Namely, the volumetric **Intersection over Union (IoU)** when ground truth meshes are available, the L1 **Chamfer Distance CD$_1$** ($\times 10^2$) and euclidean distance based **F-Score (FS)** when ground truth points are available, and finally **Normal Consistency (NC)** when ground truth normals are available. We detail the expressions of these metrics in the supplementary material.

### 4.3 Datasets

**ShapeNet** [12] consists of various instances of 13 different synthetic 3D object classes. Following [80], we use the first 20 shapes from the test split of each class for evaluation. We generate inputs of different sizes (10k, 3k, 500) while adding gaussian noise of standard deviation 0.005 following the literature (*e.g.* [7, 57]). **ScanNet v2**[18] is a challenging real dataset containing 1513 room scans as captured with an RBG-D camera. We use 100 scans from the test split for evaluation. We sample 10k points from the scans as input. **Faust** [6] consists of real scans of 10 human body identities in 10 different poses. We use the entire 100 scans for evaluation. We sample 10k points from the scans as inputs. We use the provided mesh registrations to compute IoU.

### 4.4 Object Level Reconstruction

Table 1: IoU ↑ of ShapeNet reconstruction.

|  | Conv | SAC | NP | Poco | Ours |
|---|---|---|---|---|---|
| Chair | 0,87 | 0,88 | 0,67 | 0,90 | 0,91 |
| Lamp | 0,77 | 0,75 | 0,64 | 0,80 | 0,87 |
| Table | 0,89 | 0,79 | 0,71 | 0,89 | 0,91 |
| Mean | 0,84 | 0,81 | 0,67 | 0,86 | **0,90** |

Table 2: CD$_1$ ↓ of ShapeNet reconstruction.

|  | Conv | SAC | NP | Poco | Ours |
|---|---|---|---|---|---|
| Chair | 0,41 | 0,39 | 0,56 | 0,42 | 0,34 |
| Lamp | 0,47 | 0,44 | 0,72 | 0,56 | 0,42 |
| Table | 0,35 | 0,56 | 0,62 | 0,38 | 0,30 |
| Mean | 0,41 | 0,46 | 0,63 | 0,45 | **0,35** |

Table 3: FS ↑ of ShapeNet reconstruction.

|  | Conv | SAC | NP | Poco | Ours |
|---|---|---|---|---|---|
| Chair | 0,95 | 0,98 | 0,87 | 0,98 | 0,99 |
| Lamp | 0,92 | 0,94 | 0,83 | 0,95 | 0,96 |
| Table | 0,98 | 0,92 | 0,80 | 0,98 | 0,99 |
| Mean | 0,95 | 0,95 | 0,83 | 0,97 | **0,98** |

Table 4: NC ↑ of ShapeNet reconstruction.

|  | Conv | SAC | NP | Poco | Ours |
|---|---|---|---|---|---|
| Chair | 0,95 | 0,95 | 0,88 | 0,95 | 0,96 |
| Lamp | 0,9 | 0,89 | 0,90 | 0,90 | 0,92 |
| Table | 0,96 | 0,93 | 0,78 | 0,96 | 0,97 |
| Mean | 0,94 | 0,92 | 0,85 | 0,94 | **0,95** |

We show here reconstruction results from 10k sized noisy point clouds of classes Chair, Table and Lamp of ShapeNet. Convolutional occupancy models were trained with 3k sized noisy inputs. We report a comparison on IoU in Table 1, CD$_1$ in Table 2, FS in Table 3, and NC in Table 4. Figures 2 and 4 shows qualitative comparisons. Our method improves consistently across all metrics over its baseline Poco, as well as the other strategies. Methods that benefit from deep data priors (Conv, SAC, Poco, Ours) outperform data prior free *overfitting* (NP). We note also that these results show that the finetuning strategy SAC is not stable and does not improve consistently on its deep convolutional occupancy network Conv. This is also confirmed in the visual results in Figure2, where the finetuning can lead to breakage in the reconstruction. Our method delivers more robust reconstruction generally with less holes and artifacts. As it can be seen in the same figure, it can particularly recover from failure cases of its baseline Poco. We believe that many of the failures of the baseline Poco are due to

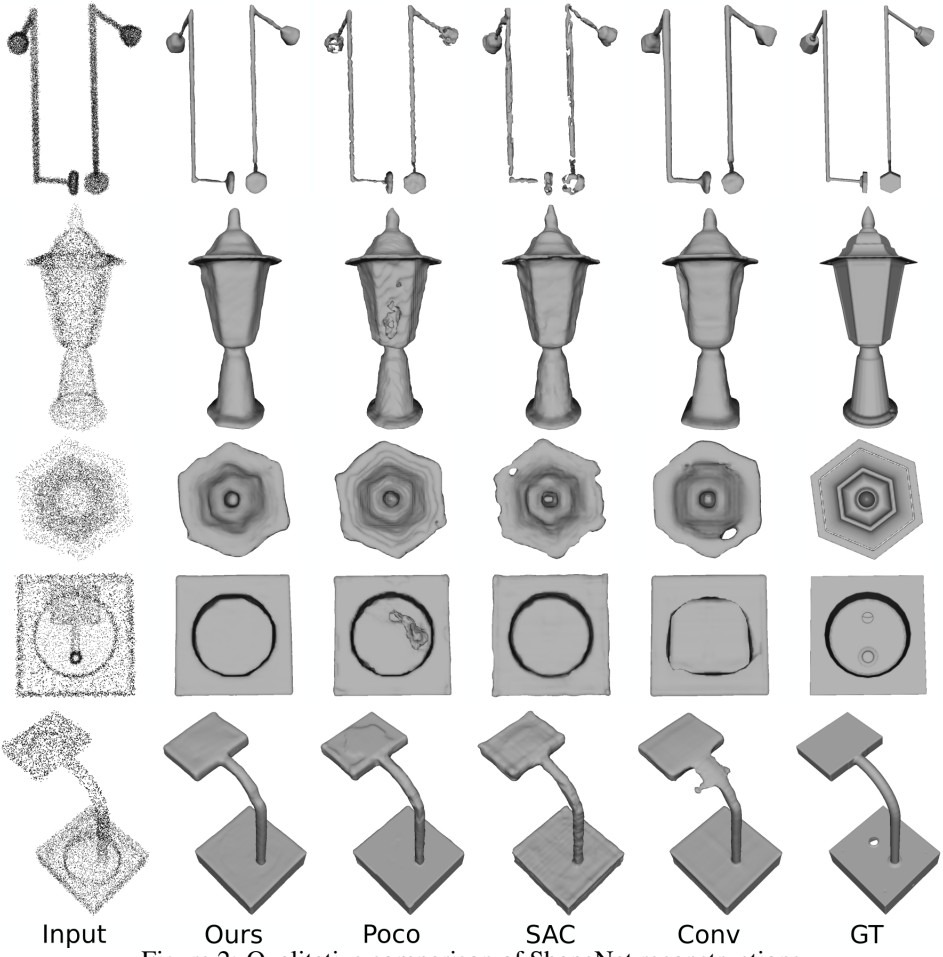

| Input | Ours | Poco | SAC | Conv | GT |

Figure 2: Qualitative comparison of ShapeNet reconstructions.

its lack of generalization w.r.t. the input size. Figure 4 shows additional visual comparisons to SPSR. We provide a numerical comparison to this classical method in the supplementary material.

## 4.5 Generalization to Real Scene Scans

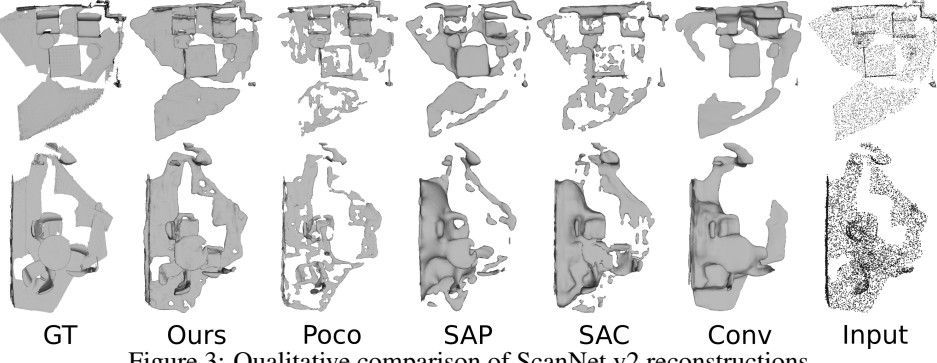

| GT | Ours | Poco | SAP | SAC | Conv | Input |

Figure 3: Qualitative comparison of ScanNet v2 reconstructions.

We evaluate here the generalization from synthetic object training to real scenes. We show reconstruction results from 10k points sampled from real scans of ScanNet v2 in Table 5. As meshes are not available we can not compute IoU here. Our method (Ours(Poco)) based on Poco outperforms the baseline and the competition, demonstrating superior generalization ability. Figure 3 shows a few

| Table 5: ScanNet reconstruction. | | | |
|---|---|---|---|
| | $CD_1\downarrow$ | NC$\uparrow$ | FS$\uparrow$ |
| SPSR | 2.27 | 0.74 | 0.68 |
| SAP | 1,12 | 0,73 | 0,67 |
| Conv | 1,22 | 0,71 | 0,60 |
| SAC | 0,97 | 0,76 | 0,77 |
| Poco | 1,11 | 0,77 | 0,79 |
| Ours (Poco) | 0,94 | **0,79** | **0,80** |
| Ours (Conv) | **0,70** | 0,77 | 0,79 |

| Table 6: Faust reconstruction. | | | | |
|---|---|---|---|---|
| | IoU$\uparrow$ | $CD_1\downarrow$ | NC$\uparrow$ | FS$\uparrow$ |
| SPSR | - | 0.48 | 0.91 | 0.91 |
| SAP | 0,90 | 0,29 | 0,93 | 0,98 |
| Conv | 0,85 | 0,41 | 0,92 | 0,95 |
| SAC | 0,77 | 0,30 | 0,92 | 0,95 |
| Poco | 0,90 | 0,32 | 0,93 | 0,97 |
| Ours | **0,92** | **0,26** | **0,95** | **0,99** |

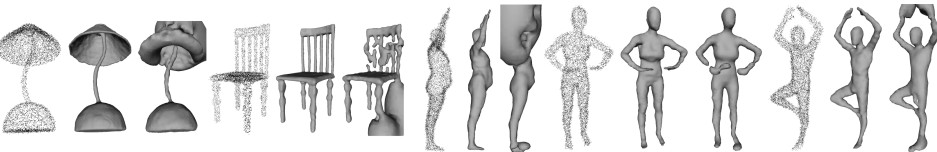

Figure 4: Qualitative comparison to Screened Poisson Surface Reconstruction (SPSR). **Input / Ours / SPSR**, in this order.

qualitative results. While we offer improved performances compared to the competition, we note that the quality of our reconstruction remains limited still is such a challenging setting. We also show that our method can be successfully applied to other existing occupancy networks, as combining our strategy with baseline Conv (Ours(Conv)) leads to a considerable improvement in its performance.

### 4.6 Generalization to Real Articulated Shape Scans

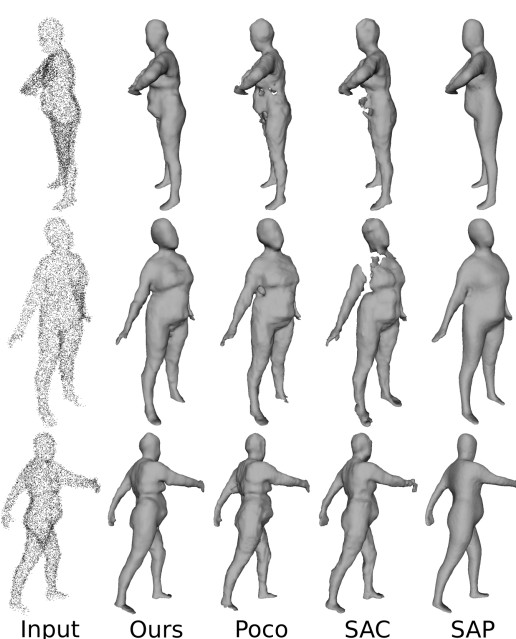

Input   Ours   Poco   SAC   SAP

Figure 5: Qualitative comparison of Faust reconstructions.

We evaluate here generalization from synthetic object training to real non-rigid human shapes. We show reconstruction results from 10k points sampled from real scans in Faust. As summarized in Table 6, both convolutional networks SAP and Poco offer good initial performances. Notice how the finetuning based SAC can improve of its baseline (Conv) $CD_1$ score while conversely worsening its IoU score. We outperform both our baseline and competing methods. Figure 5 shows a visual comparison of the results, where we recover from some failures of our baseline (Poco), and show limitations of the finetuning strategy in SAC. Figure 4 shows additional visual comparisons to SPSR.

### 4.7 Ablative Analysis

We conduct further ablative studies of our proposed method on the Lamp class of ShapeNet, as we found its shapes to be challenging and displaying many fine and complex features. Table 7 shows the performance of both our method and our baseline network Poco [7] for inputs of size 500, 3k and 10k. We note that Poco has been trained on 3k sized inputs. Notice first how the performance of Poco improves in $CD_1$ while deteriorating in IoU. This shows the performance of this model does not necessarily scale with input size. Our method on the other hand shows healthy consistent improvements over the baseline Poco, which also scale with input size. We notice also that the performance gap w.r.t. Poco is slightly tighter in the sparsest setup (500 points). We believe this is due to the inductive bias of kernel methods favouring smoothness, as such a prior can prove less effective for extremely sparse data.

Table 8 shows an ablation of the design choices of our method. Our performance drops as expected whether we disable the kernel hyperparameters $\boldsymbol{\theta}$ finetuning (w/o $\boldsymbol{\theta}$), the Nyström samples $\mathbf{Z}$ finetuning (w/o $\mathbf{Z}$). We note also the importance of learning the regression in feature space. We notice that reconstruction often fails when we attempt to solve the regression in euclidean space. Hence our w/o Feat. version uses a Fourier positional encoding to stabilize this baseline. However, this version still under performs compared to our final model. Disabling the tuning all together (w/o tune) results in a performance drop as well. Increasing the number of number of Nyström samples in this context improves the performance.

Table 8 provides also an ablation the number of Nyström samples $m$ while tuning (Ours). We find that generally the higher the number of inducing features $\mathbf{Z}$ the better the performance, as it was the case without hyper-parameter optimization (w/o tune). This is expected as the approximation error of the Nyström KRR decreases with more inducing points. Additionally, more learnable inducing points implies more representation power for the KRR function. Increasing $m$ comes however with increased computation overhead due to both solving the Nyström KRR with a higher number of inducing features $\mathbf{Z}$ and also learning these features. Increasing $m$ beyond a certain limit can also lead to overfitting. As our aim in this work is an efficient approach to improve generalization, we find a relatively small $m$ value (500) already sufficient to produce satisfactory shape predictions and offers a good performance/compute overhead trade-off.

Table 7: Ablation of input size.

|  | IoU↑ | $CD_1$↓ | NC↑ | FS↑ |
|---|---|---|---|---|
| Poco(500) | 0,71 | 0,85 | **0,86** | 0,84 |
| Ours (500) | **0,73** | **0,80** | 0,85 | **0,85** |
| Poco(3k) | 0,85 | 0,60 | 0,91 | **0,96** |
| Ours (3k) | **0,86** | **0,52** | **0,92** | **0,96** |
| Poco(10k) | 0,80 | 0,56 | 0,90 | 0,95 |
| Ours (10k) | **0,87** | **0,46** | **0,92** | **0,96** |

Table 8: Ablation of our model.

|  | IoU↑ | $CD_1$↓ | NC↑ | FS↑ |
|---|---|---|---|---|
| w/o tune ($m = 1k$) | 0.84 | 0.61 | **0.92** | 0.96 |
| w/o tune ($m = 5k$) | **0.87** | 0.55 | **0.92** | **0.97** |
| w/o tune ($m = 10k$) | **0.87** | 0.53 | **0.92** | 0.96 |
| w/o Feat. | 0.45 | 2.23 | 0.79 | 0.58 |
| w/o $\boldsymbol{\theta}$ | 0.86 | 0.52 | 0.92 | 0.95 |
| w/o $\mathbf{Z}$ | 0.86 | 0.53 | 0.92 | 0.96 |
| Ours ($m = 500$) | 0.87 | 0.42 | **0.92** | 0.96 |
| Ours ($m = 1000$) | **0.88** | **0.41** | **0.92** | **0.97** |
| Ours ($m = 5000$) | 0.86 | 0.43 | 0.91 | 0.96 |

Finally, we note that we chose to use an equal number of input and augmented samples (Equation 5) empirically as it gave the best overall results. It is possible that for high levels of noise, a higher number of augmented samples could prove beneficial. However, higher levels of noise for the input also imply less accurate predictions from the occupancy network, meaning less reliable pseudo-labels for the augmented samples.

## 5 Limitations

Extreme generalization from relatively sparse point cloud is still challenging for our method as well as the literature and we will seek further improvement in future work. Although it offers robustness, the inherent inductive bias in our method also favours smoothness, which can prove not ideal for representing the highest level of detail. Ablation of number of the NKRR Nyström samples $m$ in Table 8 shows that the KRR can possibly suffer from overfitting for large values of $m$, hence setting this hyperparameter can be challenging. We provide visualizations and a short discussion of the behavior of our method under some baseline failures in the supplementary material.

## 6 Conclusion

We presented a method for shape reconstruction from unoriented point cloud that combines the data prior of a preexisting deep reconstruction network and an efficient non-parametric interpolant. The resulting combination offers improved generalization over the network baseline, network test-time finetuning, and several state-of-the-art methods. We hope we can inspire more work in the direction of fine-tuning and transfer learning from preexisting feedforward occupancy networks, especially since there are many competing strategies that could be explored. For instance, while we propose here to improve the boundary decision by learning the shape function, an alternative strategy could be to tune solely the features instead.

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
