# Robustifying Generalizable Implicit Shape Networks with a Tunable Non-Parametric Model
## – Supplementary Material –

**Amine Ouasfi**     **Adnane Boukhayma**
Inria, Univ. Rennes, CNRS, IRISA, M2S, France

## Quantitative comparison to SPSR [3] in ShapeNet [2]

Table 1: Comparison to PCA normals + SPSR.

|      | IoU $\uparrow$ | $CD_1\downarrow$ | NC$\uparrow$ | FS$\uparrow$ |
|------|------|------|------|------|
| SPSR | 0.65 | 1.28 | 0.84 | 0.74 |
| Ours | **0.90** | **0.35** | **0.95** | **0.98** |

Table 1 shows a numerical comparison to Screened Poisson Surface Reconstruction (SPSR) [3] on classes Tables, Lamps and Chairs of ShapeNet [2]. Input point clouds are of size 10k with 0.005 standard deviation Gaussian noise. We remind that our method builds on Poco [1].

## Additional Qualitative Results for ShapeNet [2] Reconstructions

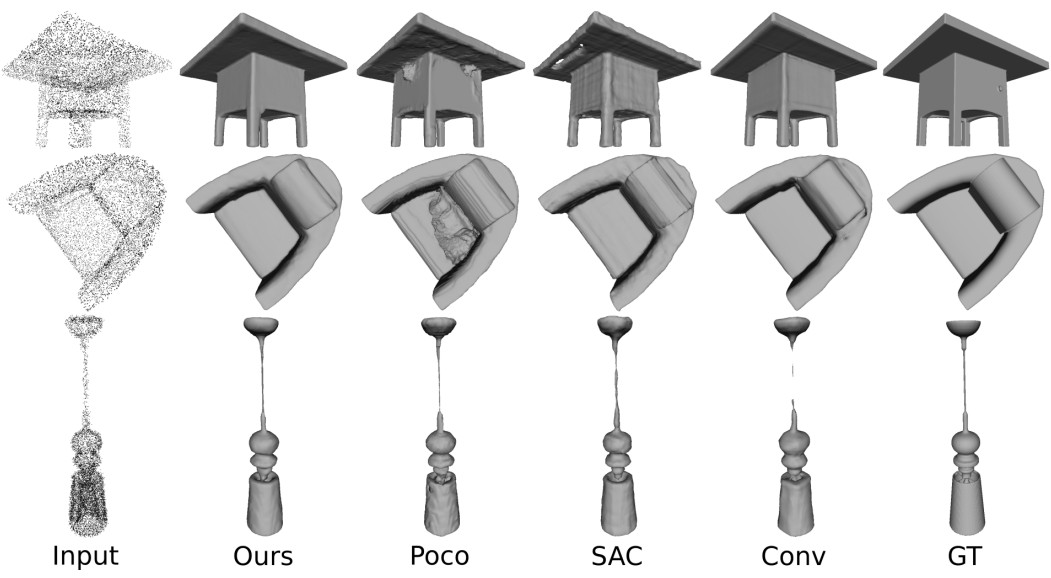

Figure 1: Additional qualitative comparison of ShapeNet reconstructions.

37th Conference on Neural Information Processing Systems (NeurIPS 2023).

Figure 1 shows additional reconstruction examples from our method, Poco [1], SAC [6] and Conv [5]. Input point clouds are of size 10k with 0.005 standard deviation Gaussian noise. While test-time finetuning (SAC) can improve on its baseline (Conv), it can also lead to worse final reconstructions. Notice how our approach can reliably improve on its beseline (Poco).

## Recovering from baseline failures

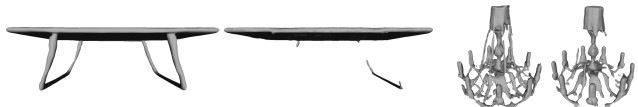

Figure 2: Comparison to our baseline in ShapeNet [2]. **Ours / Baseline**.

Figure 2 shows cases where we can recover from failures of our baseline (Poco [1]). Input point clouds are of size 10k plus a 0.005 standard deviation Gaussian noise. We improve **chamfer error** from **0.85 (baseline)** to **0.27 (Ours)** for the table example on the left, and from **1.22 (baseline)** to **0.86 (Ours)** for the chandelier on the right. As part of our limitations, and despite improving on the reconstruction of the network baseline, it is still challenging for our method to recover the finest structures and intricate details of a complex shape such as the chandelier example.

## Evaluation Metrics

Following the definitions from [1], we present here the formal definitions for the metrics that we use for evaluation in the main submission. We denote by $\mathcal{S}$ and $\hat{\mathcal{S}}$ the ground truth and predicted mesh respectively. All metrics are approximated with 100k samples from $\mathcal{S}$ and $\hat{\mathcal{S}}$.

**Intersection over Union (IoU)**    The volumetric Intersection over Union is the ratio of the intersection of the inside volumes of the meshes to their union:

$$\text{IoU} = \frac{|\ \{x \in \Omega : x \text{ inside } \mathcal{S} \text{ and } \hat{\mathcal{S}}\}\ |}{|\ \{x \in \Omega : x \text{ inside } \mathcal{S} \text{ or } \hat{\mathcal{S}}\}\ |},$$

where $|.|$ represents the sets cardinality, which is approximately determined by sampling 100k points from the bounding volume $\Omega$ of $\mathcal{S}$.

**Chamfer Distance ($\text{CD}_1$)**    The $L_1$ Chamfer distance is based on the two-ways nearest neighbor distance:

$$\text{CD}_1 = \frac{1}{2|\mathcal{S}|} \sum_{v \in \mathcal{S}} \min_{\hat{v} \in \hat{\mathcal{S}}} \|v - \hat{v}\|_2 + \frac{1}{2|\hat{\mathcal{S}}|} \sum_{\hat{v} \in \hat{\mathcal{S}}} \min_{v \in \mathcal{S}} \|\hat{v} - v\|_2.$$

**F-Score (FS)**    For a given threshold $\tau$, the F-score between the meshes $\mathcal{S}$ and $\hat{\mathcal{S}}$ is defined as:

$$\text{FS}\left(\tau, \mathcal{S}, \hat{\mathcal{S}}\right) = \frac{2\,\text{Recall} \cdot \text{Precision}}{\text{Recall} + \text{Precision}},$$

where

$$\text{Recall}\left(\tau, \mathcal{S}, \hat{\mathcal{S}}\right) = |\ \{v \in \mathcal{S}, \text{ s.t. } \min_{\hat{v} \in \hat{\mathcal{S}}} \|v - \hat{v}\|_2 \langle \tau\}\ |,$$
$$\text{Precision}\left(\tau, \mathcal{S}, \hat{\mathcal{S}}\right) = |\ \{\hat{v} \in \hat{\mathcal{S}}, \text{ s.t. } \min_{v \in \mathcal{S}} \|v - \hat{v}\|_2 \langle \tau\}\ |.$$

Following [4] and [5], we set $\tau$ to 0.01.

**Normal consistency (NC)**    We denote here by $n_v$ the normal at a point $v$ in $\mathcal{S}$. The normal consistency between two meshes $\mathcal{S}$ and $\hat{\mathcal{S}}$ is defined as:

$$\text{NC} = \frac{1}{2|\mathcal{S}|} \sum_{v \in \mathcal{S}} n_v \cdot n_{\text{closest}(v, \hat{\mathcal{S}})} + \frac{1}{2|\hat{\mathcal{S}}|} \sum_{\hat{v} \in \hat{\mathcal{S}}} n_{\hat{v}} \cdot n_{\text{closest}(\hat{v}, \mathcal{S})},$$

where
$$\text{closest}(v, \hat{\mathcal{S}}) = \text{argmin}_{\hat{v} \in \hat{\mathcal{S}}} \|v - \hat{v}\|_2.$$