# OpenReview forum: "Robustifying Generalizable Implicit Shape Networks with a Tunable Non-Parametric Model"
_NeurIPS.cc/2023/Conference — NeurIPS 2023 poster_

### Official Review · Reviewer_q2Vh · 2023-07-04

**Soundness:** 4 excellent
**Presentation:** 3 good
**Contribution:** 4 excellent
**Rating:** 5
**Confidence:** 2

**Summary:**

The paper presents a novel shape reconstruction method (point cloud-> reconstructed implicit shape) that combines a deep shape prior and a non-parametric interpolant. Existing deep shape reconstruction methods are typically trained on synthetic datasets (to obtain full supervision) and suffer from a domain shift when applied to real examples. In addition, they are sensitive to the number of points in the input point cloud. To address scaling, the paper proposes to use Kernel Ridge Regression (KRR), a non-linear interpolant that adds the condition that the points input at test time need to lie on the output implicit surface.

**Strengths:**

A novel method for implicit shape reconstruction that combines deep shape priors and implicit shape outputs is proposes. The technique is innovative and generates successful reconstructions on public benchmarks (ShapeNet, FAUST, and ScanNet) and outperforms three public baselines (Poco, Conv, SAP, SAC). Detailed ablation studies on the number of input points and KRR design choices (tuning, number of samples).

**Weaknesses:**

The key weakness of the works is that the paper could be written more clearly. The methodology section has several grammatical errors and is hard to follow. It would be helpful to have more clarity and explanations for a novice reader.

**Questions:**

The method seems to rely on a pretrained occupancy network. Does this affect the performance or expressiveness of the technique?

What is the computational time of the proposed method? From Algorithm 1, it appears to be iterative.

How sensitive is the method to the percentage of points that covers in the input, i.e., what if only a very partial view is provided?


**Limitations:**

The key contribution of the paper is that improves the synthetic to real gap, however, the few results on real data (sections 4.5 and 4.6) seems to indicate that performance is still significantly limited. It would help to identify specific example cases where the proposes method offers a significant advantage.

---

> ### Author Rebuttal · Authors · 2023-08-10
>
> ###  “The paper could be written more clearly”:
>
> We will improve the writing of the paper for the next version.
>
> ### “The method seems to rely on a pretrained occupancy network. Does this affect the performance or expressiveness of the technique”:
>
> The very objective (line #5 #39) of this work is to efficiently improve the generalization of a readily available occupancy network (typically priorly trained on large data using substantial resources). While our final performance is affected by our network baseline, we show in our experiments that we consistently outperform this baseline across all metrics and all 3 data-sets. Figure 2 in the rebuttal PDF shows an example where our baseline Poco fails while we can recover, thanks to the re-injection of the condition that inputs are at the levelset at test time. Furthermore, as requested by reviewer JKo3, we show in Table 2 in the PDF that our approach can be successfully applied to other existing occupancy networks (Conv [55]) as well besides Poco [7].
>
> ### “What is the computational time of the proposed method? From Algorithm 1, it appears to be iterative”:
>
> As reported line #213, for a 10k sized input point cloud, our method takes around 10 secs to converge on a NVIDIA RTX A6000 GPU. Other presented test time optimization competition SAC [67]: 60 secs, NP [43]: 300 secs, using their available pytorch implementations.
>
> ### “How sensitive is the method to the percentage of points that covers in the input, i.e., what if only a very partial view is provided?”:
>
> We did not address here the problem of reconstruction from incomplete or partial point cloud and we plan to do this in future work. We note that addressing this problem using our strategy would require using special generalizable occupancy networks that were trained specifically with such simulated partial input scenario.
>
> ### “The key contribution of the paper is that improves the synthetic to real gap, however, the few results on real data (sections 4.5 and 4.6) seems to indicate that performance is still significantly limited”:
>
> We improve both the gap caused by change in input density (Tables 1,2,3,4,7) and the synthetic to real gap (Tables 5,6). We outperform common state-of-the-art and improve considerably and consistently on our baseline on 3 datasets. For the most important metric i.e. Chamfer distance (CD1), we offer an improvement of 22%in ShapeNet (Table 2), 15% in ScanNet (Table 5) and 19% in Faust (Table 6) wrt to our occupancy network baseline. Figure 2 in the PDF shows additional examples where our method can recover from failures of our baseline.  We do not claim that we solve the generalization problem as it is a notoriously hard one, but we propose a novel and efficient way to approach it in the context of implicit shape reconstruction from point cloud, that also achieves very encouraging results.

---

> > ### Comment · Reviewer_q2Vh · 2023-08-15
> >
> > Thank you for the rebuttal, which addressed most of my concerns and I raised my rating.

---

### Official Review · Reviewer_Pm8Q · 2023-07-04

**Soundness:** 3 good
**Presentation:** 3 good
**Contribution:** 3 good
**Rating:** 6
**Confidence:** 4

**Summary:**

To address the poor performance of feed-forward models on noisy real data, the authors propose using test-time Kernel Ridge Regression (KRR) to fine-tune the results for implicit shape networks, specifically in the context of recovering 3D reconstruction surfaces from noisy point clouds. The method involves incorporating a feedforward network called Poco and utilizing the input point cloud and the network's approximated logits to determine the optimal KRR for representing the final implicit shape. This method improves the performance of Poco.

**Strengths:**

1. The paper introduces an additional post-processing step, utilizing information from the input test data to enhance surface reconstruction and improve generalization ability for unseen cases. The idea is novel, as it combines a non-parametric model with a neural network to enhance the performance of the latter further.

2. The authors have evaluated their method on various datasets to assess the entire pipeline.

**Weaknesses:**

1. The paper does not mention the overhead of the current method. Since the KRR model needs to be optimized for different point cloud inputs, it is crucial to report the optimization time in this paper.

2. Currently, the training samples for KRR consist of an equal number of noisy input and network-predicted samples. Could the authors explain the reasoning behind this decision? It would be helpful to understand why they did not include more or fewer network-predicted samples. In cases where the input data is highly noisy, the inclusion of more network-predicted samples might prove more beneficial.

3. The number of Nystrom Samples, denoted as "m," should have an impact on the final output performance. However, the paper lacks a clear discussion on this aspect. In Table 8, there is an option labeled "without tune" with different values of "m." Does this mean that there was no optimization performed on theta and Z? It would be better to include a comparison with optimization and different values of "m" for a comprehensive analysis.

4. Some typos: broken reference in line 41; Eq. 5, for the input point cloud, the logit should be 0.5, on the surface?

**Questions:**

See weakness

**Limitations:**

The

---

> ### Author Rebuttal · Authors · 2023-08-10
>
> ### “1. The paper does not mention the overhead of the current method. Since the KRR model needs to be optimized for different point cloud inputs, it is crucial to report the optimization time in this paper”:
>
> As reported line #213, for a 10k sized input point cloud, our method takes around 10 secs to converge on a NVIDIA RTX A6000 GPU. Other presented test time optimization methods SAC [67]: 60 secs, NP [43]: 300 secs, using their available pytorch implementations.
>
> ### “2. Currently, the training samples for KRR consist of an equal number of noisy input and network-predicted samples. Could the authors explain the reasoning behind this decision? It would be helpful to understand why they did not include more or fewer network-predicted samples. In cases where the input data is highly noisy, the inclusion of more network-predicted samples might prove more beneficial”:
>
> We chose this ratio empirically as it gave us the best overall results. We note that we experimented in this paper within the standard reconstruction benchmarks . It is possible indeed that for even higher levels of noise, a higher number of augmented samples could prove beneficial. However, higher levels of noise for the input also imply less accurate predictions from the occupancy network, meaning less reliable pseudo-labels for the augmented samples. We will include a discussion in the next version.
>
> ### “3. The number of Nystrom Samples, denoted as "m," should have an impact on the final output performance. However, the paper lacks a clear discussion on this aspect. In Table 8, there is an option labeled "without tune" with different values of "m." Does this mean that there was no optimization performed on theta and Z? It would be better to include a comparison with optimization and different values of "m" for a comprehensive analysis.”
>
> Following the request of the reviewer we provide an ablation of parameter “m” with optimization in the rebuttal PDF in Table 3. We find that the higher the number of inducing features Z the better the performance, as it was the case without hyper-parameter optimization (Table 8 w/o tune). This is an expected result as the approximation error of the Nyström KRR decreases with more inducing points. On top of that, more learnable inducing points implies more representation power for the KRR function. Increasing m comes however with increased computation overhead due to both solving the Nyström KRR with a higher number of inducing features Z and also learning these features. Increasing the number of learnable inducing features beyond a certain limit can also lead to over-fitting. As our aim in this work is an efficient approach to improving generalization, we find a relatively small m value (500, line #221) already sufficient to produce satisfactory shape prediction results and offers a good performance/compute overhead trade-off. We will include the ablation table and a thorough discussion of this point in the next version.

---

> > ### Comment · Reviewer_Pm8Q · 2023-08-15
> >
> > Thanks for the rebuttal and it resolves some of my concerns. I will maintain the same rating.

---

### Official Review · Reviewer_7xQW · 2023-07-05

**Soundness:** 3 good
**Presentation:** 2 fair
**Contribution:** 2 fair
**Rating:** 5
**Confidence:** 3

**Summary:**

The paper proposes a method for improving shape reconstruction from a set of sparse and noisy unoriented point clouds. As data-driven trained networks can have failures on out-of-distribution examples, this method proposes to take the input point cloud, the predicted surface points, and the features generated by the pre-trained network, and estimate a non-parametric set of kernels to represent the output shape occupancy. Results on shape net, Faust, and scannet show improved reconstruction performance compared to prior deep-network only approaches, and recent work that perform test-time fine-tuning.

**Strengths:**

- [S1] The method outperforms prior work it compares in SoTA deep network shape prediction and reconstructs well thin surfaces. It was nice to see strong generalization results on human shape reconstruction trained only from shapenet data.
- [S2] The idea of inducing additional priors and structure in shape prediction is a nice direction to generalization

**Weaknesses:**

- [W1] The related work covers shape representations, optimization-based reconstruction, and feed-forward prediction (either supervised or unsupervised). No context is provided of how this work fits into the field. See Q1 below, the paper needs more motivation and why this particular approach of post-processing using a pre-trained network is a fruitful direction to pursue for improving reconstruction from sparse and noisy point clouds.
- [W2] The paper is missing a baseline to classical methods such as PCA-neighbor-based normal estimation + screened-poisson surface reconstruction, using input points and optional pseudo-points from the network prediction.
- [W3]  No qualitative visuals of failures provided, or discussion of other limitations, such as the assumption that the input point cloud is a contiguous surface and that kernel prior of locally smooth surfaces doesn’t always hold (not just recovering details) - how does the reconstruction perform on an input of several nearby rocks with jagged surfaces and holes?  I wish the paper also showed more qualitative examples - only one from scannet and three from Faust are shown.

Minor weakness:
- Multiple typos, for example section 3.1: (L136 isntance —> instance, L138 contenting —> contending, L148 and —> an, L153 order-invariantly —> pooled to be order invariant, L154 “Without loss of generality we build in this work on the model “).
- Writing style - some parts of the text felt extraneous. If the method focuses on post-processing, why devote L136-L147 talking about different input network architecture designs. It seems the proposed method can work with most pretrained-network shape reconstruction methods, as long as it can provide pseudo-labels and features about the 3D space points.

**Questions:**

[Q1] Why is robustifying pre-trained shape reconstruction networks the desired approach to improving shape reconstruction, instead of directly designing a reconstruction method with kernels [1]?  What benefit does this alternative direction offers - why post-correct rather than inducing these priors directly in the representation and in the original training. What is the main novelty and contributions over neural kernel fields, and neural kernel surface reconstruction?
[1] Neural Kernel Surface Reconstruction CVPR 2023 (and its predecessor neural kernel fields CVPR 2022). The related work cites NKF in “Reconstruction from Point Cloud,” but does not describe how the submitted paper fits in this context.
The introduction lacks motivation other than describing the limitations of existing deep-learning based shape reconstruction methods.


**Limitations:**

See  W3, while they are mentioned, I feel limitations was not adequately addressed with examples. For a visual topic like shape reconstruction, examples would help better illustrate them.

---

> ### Author Rebuttal · Authors · 2023-08-10
>
> ### “Q1/W1. Why is robustifying pre-trained shape reconstruction networks the desired approach to improving shape reconstruction, instead of directly designing a reconstruction method with kernels ? What benefit does this alternative direction offers - why post-correct rather than inducing these priors directly in the representation and in the original training”:
>
> - While building generalizable occupancy networks including kernel priors is definitely a promising direction, one might not have the important resources, the time and data required to build such models. Alternatively, given readily available generalizable shape models, we propose here an efficient solution to improve their generalization at test time instead. We believe both directions to be practical and useful for the community.
>
> - Rather than using a simple KRR, depending on manually crafted fixed kernels, we explore in this work the hyper-parameter space of Nystrom-KRR in the context of shape reconstruction by learning the inducing points  and the kernel parameters. Consequently we adapt our hypothesis space in accordance to the shape we aim to recover. It is not straightforward to integrate this process within the training of a generalizable network, and this is a challenge we are looking forward to uncover as part of future work.
>
> ### “What is the main novelty and contributions over neural kernel fields, and neural kernel surface reconstruction?”:
>
> CVPR 23 Proceedings were not available yet at the NeurIPS 23 deadline. Existing learning based kernel reconstruction methods ([78] Neural splines CVPR 2021, [76] Neural kernel fields CVPR 2022) differ from our work in the following:
> - Both [76] and [78] address the problem of reconstruction from points and their normals. Differently, we use KRR to reconstruct from points only which is a much harder problem.
> - Differently from [76] and [78],  we propose a strategy to reduce the generalization gap by  solving our regularized ERM problem (Equation 6)  in a RHKS  space  adapted to  the shape we would like to recover. This is possible thanks to the unique correspondence between RKHS and kernels. Hence, instead of relying on handmade fixed kernels as in  [76] and [78], we learn the kernel parameters using a loss function that avoids overfitting on the data  term (Equation 10). Additionally, we can  dynamically adjust this function within the reproducing kernel Hilbert space by learning the Nystrom  samples. This is a novel approch that was not explored in the context of shape reconstruction , as the  resulting function from this optimization is more complex and expressive then a simple KRR mapping   [76],78]. We will clarify further relation to [76] and [78] in our related work section.
>
>
> ### “[W2] The paper is missing a baseline to classical methods such as PCA-neighbor-based normal estimation + screened-poisson surface reconstruction”:
> Following the request of the reviewer, we provide an additional classical baseline (PCA normals + SPSR) in the rebuttal PDF (Figure 1, Table 1). We provide quantitative and qualitative comparison. This baseline deteriorates under low input point density and due to the inaccuracies of nearest neighbor based normal approximation.
>
> ### “[W3] No qualitative visuals of failures provided, or discussion of other limitations, such as the assumption that the input point cloud is a contiguous surface and that kernel prior of locally smooth surfaces doesn’t always hold (not just recovering details) - how does the reconstruction perform on an input of several nearby rocks with jagged surfaces and holes?”:
> We addressed in this work reconstruction from uniform point cloud as it is the case in our competition work. We defer more challenging inputs such as partial, incomplete and extremely noisy point clouds to future work. We note that while we restrain our hypothesis space to a RHKS,  we ensure this space to be rich enough to approximate any continuous function arbitrarily well by using a gaussian kernel. Learning the hyperparemters of the Nystrom KRR including the kernel parameters and the Nystrom samples  allows as to solve the KRR in a RKHS specialized to the  shape we aim to recover. In addition,  we can dynamically adjust inputs (Nystrom samples) in case they are noisy or not placed conveniently initially wrt the underlying surface. As mentioned in  line #287 as part of limitations, kernels do favor smoothness which can prove tricky for recovering highly detailed shapes. We focus in this work on the robustness of the reconstruction rather than finer levels of detail.
> We provide additional limitation in the rebuttal PDF and we will discuss them further in the next version. First ablation of number of nystrom samples m in Table3 shows that the KRR can possibly suffer from overfitting for large values of m, hence the choice of this hyperparameter can be challenging. Additionally, Figure 2 show an example where, despite improving on the reconstruction of the network baseline, it is still challenging for our method to recover the finest structures and intricate details of a complex shape.
>
> ### “I wish the paper also showed more qualitative examples - only one from scannet and three from Faust are shown”:
> We provide additional reconstructions in the supplementary material and the Rebuttal PDF.

---

> > ### Comment · Reviewer_7xQW · 2023-08-16
> >
> > Thank you to the authors for the rebuttal.
> >
> > "Rather than using a simple KRR, depending on manually crafted fixed kernels" - the kernels in NKSR (concurrent work) are not that manual - its just a dot product of the features which is fully data-driven + bezier interpolation. There's not much hyperparameters involved - all parameters belong in the feature generation from points with the MLP. So I disagree with this statement, but maybe I misunderstood something. Nystrom sampling is interesting, but space partitioning like in NKSR (concurrent work) to overcome this complexity seems to be a more promising direction.
> >
> > "Differently, we use KRR to reconstruct from points only which is a much harder problem"
> > while [76] requires normals, one could apply pre-processing on the point cloud to estimate normals. Additionally the proposed method is evaluated on quite dense point clouds that are not noisy, so normal estimation with PCA could be quite good.
> >
> > Regarding W3, I didn't mention partial, incomplete and noisy point clouds as failures - I asked about locally non-smooth surfaces that have holes as part of their actual geometry, not the observation (like multiple rocks). And as I mentioned, its not just about recovering details, but about correctly approximating the topology. I would have hoped that a visual could showcase this in the rebuttal.
> >
> > But overall I appreciate the rebuttal, the clarification of this work's differences to prior work that leverage kernels in reconstruction, and the classical baseline comparison. I raise my rating to weak accept, if the revised paper can have an updated related work, improve its motivation in the intro, and fix the multiple typos throughout the paper.  And especially please include more visuals in the main paper and supplementary, it gives readers more insight into the method.  Thank you

---

> > > ### Author Response · Authors · 2023-08-17
> > >
> > > We thank the reviewer for his valuable feedback.
> > >
> > > We agree that the joint learning of features, along with space partitioning as seen in the concurrent work NKSR (CVPR23), represents a promising avenue for training generalizable feedforward models. The choice of the kernel in NKSR (dot product of features) induces an RKHS of linear functions in feature space such that **during training** the encoder learns a feature representation adapted to this choice of “decoders”.
> > >
> > > For test-time adaption of a pretrained network, the existing features are not necessarily adapted to linear decoders. Hence, we opted for a richer RKHS, while ensuring a **shape specific trade-of between expressiveness and regularization** thanks to the tuning of the N-KRR hyper-parameters. For large scale data, space partitioning at test-time is an interesting future direction. We will include concurrent work NKSR in our related work section.
> > >
> > > The updated version of our paper includes the following changes: Introduction updated with **motivation** of improving on existing networks, motivation of doing so with a learnable N-KRR, and motivation of using a gaussian kernel. **Related work** updated with clear differentiation from [76,78], introduction of concurrent work NKRS, and clarification of weaknesses of some prior work. Figure 1 updated to be more intuitive. Method section updated to be clearer. Results updated with **comparison to SPSR**. Implementation details updated with justification of choice of number of augmented points. Ablations updated with analysis of “m” under optimization, and generalizability: using Conv[55] as a baseline for our method. Limitations updated with discussion of the effect of choice of “m”, and behavior under failures of the baseline. **More visuals** were added to the paper and supplementary, typos were fixed, and writing was improved overall.
> > >
> > > We would have been very happy to provide our revised version. Unfortunately, it seems that as of this year it is no longer possible to upload a revision during discussion: (https://neurips.cc/Conferences/2023/PaperInformation/NeurIPS-FAQ “Can we upload a revision of our paper during the rebuttal/discussion period? No revisions are allowed until the camera-ready stage.“)
> > >
> > > We are sorry about this and we thank you for your understanding.

---

### Official Review · Reviewer_6hDU · 2023-07-05

**Soundness:** 3 good
**Presentation:** 2 fair
**Contribution:** 2 fair
**Rating:** 5
**Confidence:** 4

**Summary:**

This paper aims at the task of implicit surface reconstruction from noisy point clouds. It tackles the model generalizability and robustness by introducing a kernel ridge regression as a implicit function that maps a shape feature to occupancy.


**Strengths:**


The proposed method achieve higher accuracy compared to prior surface reconstruction method. It shows certain robustness when applied in real datasets.


**Weaknesses:**


a. What is the motivation behind KRR? How does it help the generalizability of the models? Why it is fitted for surface reconstruction from noisy point clouds? The authors should demonstrate the intuition behind the introduction of KRR, compared to other occupancy decoders.
b. From the results on ShapeNet, we see that even for test dataset that have the same density and synthetic/real property as the training dataset, the proposed method still outperforms other methods with a notable margin. Why is that?
c. The writing is not very clear, especially how the introduced KRR is integrated in a deep learning network. Is the feature encoder fixed in the proposed method? Is the optimization of KRR conducted for each test sample? If so, the experiments need to compare the test time and accuray simultaneously. Also I have confusion on the discussion from L136-L147. How the intrinsity or extrinsity of point feature could effect the proposed method?  Besides, Fig. 1 is not organized well. There too many notations and unclear text color, which does not help understanding.

Typos:
L42: Wrong reference
L45: ; -> ,
L136: isntance -> instance
L148: and MLP -> an MLP


**Questions:**

How the ground truth point clouds of the real dataset (ScanNet v2 and Faust) acquired? What is the difference between ground truth point clouds and noisy point clouds for these real datasets. Are the noise introduced during scanning or added afterwards additionally?


**Limitations:**

The authors have addressed the limitations.

---

> ### Author Rebuttal · Authors · 2023-08-10
>
> ### "a.What is the motivation behind KRR? How does it help the generalizability of the models? Why it is fitted for surface reconstruction from noisy point clouds? The authors should demonstrate the intuition behind the introduction of KRR, compared to other occupancy decoders."
>
> While feedforward generalizable occupancy networks can offer very good performances and fast inference, they can underfit the input point cloud as the knowledge that those samples are on the surface is not explicitly enforced. One solution to improve this aspect (line #39) and hence the generalization of these networks at test time, is to fine tune the network as in SAC [7] with such a loss. We find this strategy to be unstable and can even lead to worst results than the baseline in many cases (see Results section). One of the reasons for this is the difficulties arising from optimizing a large number of neural network parameters.
>
> These instabilities  highlight the necessity for regularization in this adaptation process. We need to balance  the need to maintain the expressive capacity of the learned functions while ensuring stability and robustness. To this end we devise three important design choices.
> Firstly, we restrain the hypothesis space of the shape functions to be a reproducing kernel Hilbert space. (RKHS).  By the Representer Theorem, the minimizer of our regularized empirical risk minimization (ERM) problem (equation 6 in the paper)  emerges naturally as the solution of a KRR problem.
> Secondly, by using a Gaussian kernel we benefit from the universality properties of the associated (RKHS) ie. a hypothesis space rich enough to approximate any continuous function arbitrarily well.
> Thirdly,  we propose a strategy to solve our regularized ERM problem (Equation 6)  in a RHKS  space  adapted to  the shape we would like to recover. This is possible thanks to the unique  correspondence between RKHS and kernels. Hence, instead of relying on handmade fixed kernels, we learn the kernel parameters using a loss function that avoids overfitting on the data term (Equation 10).
>
> Thus, the primary motivation behind KRR  is to  avoid  the difficulties arising from optimizing a large number of neural network parameters, as in SAC, while maintaining the expressive capacity necessary for effective shape modeling.
>
>
> ### “b. From the results on ShapeNet, we see that even for test dataset that have the same density and synthetic/real property as the training dataset, the proposed method still outperforms other methods with a notable margin. Why is that?”:
>
> As it has been noted by previous work too [76,7], feedforward generalizable occupancy networks can underfit the input point cloud, i.e. there is no explicit guaranty that input points are at the predicted shape function level set from a single forward pass. Enforcing this constraint under well chosen regularizations (i.e. kernel prior) leads to improvements at test time.
>
> ### “c. The writing is not very clear, especially how the introduced KRR is integrated in a deep learning network”:
>
> We will clarify these details and improve the writing in the next version.
>
> ### “c. Is the feature encoder fixed in the proposed method?”:
> Yes.
>
> ### “c. Is the optimization of KRR conducted for each test sample? If so, the experiments need to compare the test time and accuracy simultaneously.”
>
> Yes, as it the case for test time optimization competition presented in the paper (e.g. SAC, NP). As reported line #213, for a 10k sized input point cloud, our method takes around 10 secs to converge on a NVIDIA RTX A6000 GPU. Other presented test time optimization methods SAC [67]: 60 secs, NP [43]: 300 secs, using their available pytorch implementations.
>
> ### “c. How the intrinsity or extrinsity of point feature could effect the proposed method?”:
>
> We build here on a method with an intrinsic encoder (Poco [7]) as it offers good performance across many benchmarks. We show in Table 2 in the rebuttal PDF that our approach can be applied successfully to a method with an extrinsic encoder as well (Conv [55]).
>
> ### “c. Fig. 1 is not organized well.”
>
> We will improve the figure, and explain the color coding more thoroughly.

---

> > ### Comment · Reviewer_6hDU · 2023-08-16
> >
> > Thank you for the rebuttal. It addresses my main concern on the motivation of KRR. I would raise my rating to borderline accept.

---

### Official Review · Reviewer_JKo3 · 2023-07-07

**Soundness:** 3 good
**Presentation:** 4 excellent
**Contribution:** 2 fair
**Rating:** 6
**Confidence:** 2

**Summary:**

This paper proposes an efficient mechanism to address the generalization issues often encountered by generalizable feedforward models trained for implicit shape reconstruction from unoriented point clouds. The solution proposed in this work is to use kernel ridge regression in the network's feature space, tuned by gradient descent. At a high-level, this approach utilizes a pre-trained shape reconstruction network to produce pseudo-labels to provide more supervision. The effectiveness of the proposed method is demonstrated on multiple datasets, including synthetic and real data.

**Strengths:**

1. The overall quality of this paper is high. The sections are well-written and cohesively presented, ensuring a smooth flow of information.
2. The extensive experiments effectively highlight the underfitting and overfitting issues that this research seeks to address. The motivation for the study is convincingly articulated.
3. The decision to implement kernel regression in the feature space of a pre-trained occupancy network is sound and logical. Even though the pre-trained occupancy network is already robust and strong, the approach taken in this paper is understandable given its objective to create a test-time augmentation method.


**Weaknesses:**

1. Comparative Analysis: Given that the proposed method is primarily a test-time optimization technique, it would be beneficial to include a comparison with more test-time augmentation methods. While the authors do mention a test-time fine-tuning baseline in lines 206-207, it might be worth investigating if there are other advanced test-time optimization techniques, specifically ones that could directly apply to Poco features.

2. Demonstrating Generalizability: The results showcased in this paper are primarily based on the Poco network, which already significantly outperforms other baseline methods as demonstrated in Table 1. Although I believe this method could be applied to other feature extractors, it would strengthen the paper to present more numerical results to support this.

**Questions:**

1. What does the k_theta in L184 represent?

**Limitations:**

The paper has discussed its limitations.

---

> ### Author Rebuttal · Authors · 2023-08-10
>
> ### “What does the k_theta in L184 represent?”
>
> As reported line #217,  k_theta is the Gaussian kernel with a single length-scale for each data-dimension in our experiments. We will clarify this. We note that by using a Gaussian kernel we benefit from the universality properties of the associated (RKHS) i.e. a hypothesis space rich enough to approximate any continuous function arbitrarily well.
>
> ### "W1. Comparative Analysis: Given that the proposed method is primarily a test-time optimization technique, it would be beneficial to include a comparison with more test-time augmentation methods. While the authors do mention a test-time fine-tuning baseline in lines 206-207, it might be worth investigating if there are other advanced test-time optimization techniques, specifically ones that could directly apply to Poco features."
>
> Apart from the work in SAC[7] and ours, we are not aware of the existence of other work in the area of fine-tuning or transfer learning from preexisting feedforward occupancy networks. We hope we can inspire more work in this direction, especially since there are many competing strategies that could be explored. For instance, while we propose here to improve the boundary decision by learning the shape function, an alternative strategy could be to tune solely the features. It's important to note that such a feature-based strategy might prove computationally expansive for large point cloud networks.
>
> ### "W2. Demonstrating Generalizability: The results showcased in this paper are primarily based on the Poco network, which already significantly outperforms other baseline methods as demonstrated in Table 1. Although I believe this method could be applied to other feature extractors, it would strengthen the paper to present more numerical results to support this."
>
> Our method can be successfully applied to other existing generalizable occupancy networks. To support this, we show the result of combining the Shapenet trained network in Conv [55] with our strategy in Table 2 in the rebuttal PDF. We perform this experiment in the ScanNet benchmark as it is the most challenging in terms of generalization. Our Conv [55] based method outperforms its baseline. We will add this result in our next version.

---

> > ### Comment · Reviewer_JKo3 · 2023-08-16
> >
> > Thanks for the rebuttal. It addresses my concerns and I would like to raise my score to weak accept.

---

### Official Review · Reviewer_LSLk · 2023-07-22

**Soundness:** 3 good
**Presentation:** 2 fair
**Contribution:** 2 fair
**Rating:** 5
**Confidence:** 4

**Summary:**

This paper aims to reconstruct smooth and high-quality surface from noisy point cloud. While previous works like Poisson surface reconstruction can extract surface from point cloud, but it's sensitive to input noise. The existing learning-based methods are able to handle noisy point cloud, but often suffer from generalizing to other dataset, real-world scenes, and large scale point cloud. The authors propose to leverage pretrained learning-based occupancy network (POCO) for extracting point cloud features and generate pseudo label for new points. The Kernel Ridge Regression (KRR) and the Nystrom samples are jointly optimized with gradient descent, representing the final shapes from the input point cloud. The experiments show that proposed method achieve better evaluation metrics (CD, IoU, NC, FS) on synthetic ShapeNet, real-world dataset ScanNet v2. The qualitative results and ablation study are conducted to verify the design choices.

**Strengths:**

- The quantitative results in Table 1-6 demonstrate that proposed method achieve better evaluation metric, including Chamfer distance, IoU, normal consistency, and F-sore. These metrics indicate the reconstructed meshes have superior quality.
- The qualitative results in the figures show that the reconstructed surface is smoother and more complete, in contrast to the baseline methods.
- The proposed method can generalize to synthetic and real-world large scale input point cloud.

**Weaknesses:**

- Formulating implicit surface with kernel-based method is not a new idea. The fundamental difference between proposed method and previous works "Neural Kernel surface reconstruction", "Neural Fields as Learnable Kernels for 3D Reconstruction" is not described in the paper, and the comparison is also missing in the experiments section, making it difficult to evaluate the technical contribution.
- The propose method relies on pretrained POCO to generate point features and pseudo points label. The authors argue that the learning-based methods suffer from generalizing to out-of-distribution data. As a result, the proposed method is likely to fail for the data that POCO cannot produce good enough initialization (e.g. incorrect occupancy prediction). As a result, the capability of generalization of proposed method does not outperform previous learning-based method significantly.
- Per-scene optimization is required to get KRR hyper-parameters and Nystrom sample points. While the optimization doesn't take much time, the comparison with baseline methods might be unfair since most of them only need single forward pass. In addition, according to the quantitative results in the tables, the number only improve slightly from POCO baseline.
- The weakness of previous works and difference of proposed method is not clear in the related works section. To be more specific, in the subsection "Unsupervised Implicit Neural Reconstruction", various previous works are mentioned along with some technical details, but the description about the their common limitation is missing, the only text that describe this group of works is "All of the aforementioned work benefits from efficient gradient computation through back-propagation in the neural network." which does not contain much information.






**Questions:**

- What's the fundamental difference between the proposed method and previous kernel-based neural surface reconstruction (see the references in the limitation section)?
- How could the optimization recover the shape surface from point cloud if the POCO backbone is not able to produce reasonable prediction?
- Is there any intuition behind the design choice where the authors decide to represent the surface as occupancy instead of signed distance function?

### Other comments
- Missing section reference in line 42
- typo in line 148
- Missing bracket in eq. 12

**Limitations:**

There's no potential negative societal impact of this paper.

---

> ### Author Rebuttal · Authors · 2023-08-10
>
> ### “What's the fundamental difference between the proposed method and previous kernel-based neural surface reconstruction?”:
>
> CVPR 23 Proceedings were not available yet at the Neurips 23 deadline. Existing learning based kernel reconstruction methods ([78] Neural splines CVPR 2021, [76] Neural kernel fields CVPR 2022) differ from our work in the following:
> - Both [76] and [78] address the problem of reconstruction from points and their normals. Differently, we use KRR to reconstruct from points only which is a much harder problem.
> - Differently from [76] and [78],  we propose a strategy to reduce the generalization gap by  solving our regularized ERM problem (Equation 6)  in a RHKS  space  adapted to  the shape we would like to recover. This is possible thanks to the unique correspondence between RKHS and kernels. Hence, instead of relying on handmade fixed kernels as in  [76] and [78], we learn the kernel parameters using   loss function that avoids overfitting on the data  term (Equation 10). Additionally, we can  dynamically adjust this function within the reproducing kernel Hilbert space by learning the Nystrom  samples. This is a novel approach that was not explored in the context of shape reconstruction , as the  resulting function from this optimization is more complex and expressive then a simple KRR mapping   [76],78]. We will clarify further relation to [76] and [78] in our related work section.
>
> ### “How could the optimization recover the shape surface from point cloud if the POCO backbone is not able to produce reasonable prediction?”:
>
> Feedforward generalizable occupancy networks such as Poco can underfit the input point cloud as the knowledge that input samples are on the surface is not explicitly enforced. One solution to improve the prediction of these networks is to ensure the input points coincide with the level set of the predicted shape function. One existing way to do this is finetuning the network as in SAC [7] with such a loss. We find this strategy to be unstable. It can even lead to worst results than the baseline in many cases (see Results section). One of the reasons for this is the difficulty of optimizing a large number of neural network parameters. Alternatively (Line #173), we propose to optimize less parameters under more constraints which yields more robustness, as our learnable KRR offers both flexibility and spatial regularization to the shape function. In the rebuttal PDF, Figure 2, we show examples where the network baseline fails while we can recover good reconstructions.
>
> ### “Is there any intuition behind the design choice where the authors decide to represent the surface as occupancy instead of signed distance function?”:
>
> This choice was based on the fact that the most common generalizable feedforward implicit reconstruction from point cloud networks (e.g. Conv[55], Poco[7]) use the occupancy representation.
>
> ### “Per-scene optimization is required to get KRR hyper-parameters and Nystrom sample points. While the optimization doesn't take much time, the comparison with baseline methods might be unfair since most of them only need single forward pass”:
>
> We compare to both feedforward baselines, and test-time optimization methods such as ours, and we outperform them all. Optimization competition includes the closest existing method to ours, based on finetuning a generalizable occupancy network at test time SAC [7], and a data prior free sota method NP [43].
>
> ### “In addition, according to the quantitative results in the tables, the number only improve slightly from POCO baseline”:
>
> We beg to differ. We find our improvements over our baseline to be important and consistent across all 3 datasets and metrics. For the most important metric i.e. Chamfer distance (CD1), we offer an improvement of 22% in ShapeNet (Table 2), 15% in ScanNet (Table 5) and 19% in Faust (Table 6).
>
> ### “The weakness of previous works and difference of proposed method is not clear in the related works section”:
> We will improve the related work discussion in the next version.

---

> > ### Comment · Reviewer_LSLk · 2023-08-16
> >
> > I appreciate the efforts made by the authors in the rebuttal process.
> > While the motivation and major difference with previous kernel-based surface reconstruction works are unclear in the original paper, the rebuttal contents clarify these points better. The authors are strongly encouraged to improve the writing in the introduction, related works, and method sections.
> > Figure 2 in the rebuttal PDF showcases some examples that baseline methods cannot do well but the proposed method does, which addresses my concern in the second point. It would be great if the authors could mention the hypothesis and evidence of this improvement in the main paper.
> > While the Chamfer distance is improved significantly, the other metrics (e.g. normal consistency, F-score) have marginal improvement.
> > Overall, the authors address my concerns about the difference and improvements with previous works, but the writing must be reorganized to enhance the clarity of motivation and technical contribution. Considering the efforts made by the authors, I would like to raise my rating to "border line accept".

---

### Author Rebuttal · Authors · 2023-08-10

We thank the reviewers for their valuable feedback. We will improve writing in the next version and correct typos pointed out by reviews. We will add the additional discussions and results provided in the rebuttal. We address the questions separately in the respective rebuttals and we summarize also bellow points that were important or common between several rebuttals. We provide the corresponding figures and tables in the PDF file.

### Motivation and Comparison to existing kernel reconstruction methods [76,78]

Our objective is to reconstruct the shape  by dynamically adjusting the network to the input point-cloud during testing. The instability observed  in SAC [7]  highlights the necessity for regularization in this adaptation process. We need to balance  the need to maintain the expressive capacity of the learned functions while ensuring stability and robustness. To this end we devise three important design choices.

Firstly, we restrain the hypothesis space of the shape functions to be a reproducing kernel Hilbert space. (RKHS).  By the Representer Theorem, the minimizer of our regularized empirical risk minimization (ERM) problem (equation 6 in the paper)  emerges naturally as the solution of a KRR problem.

Secondly, by using a Gaussian kernel we benefit from the universality properties of the associated (RKHS) ie. a hypothesis space rich enough to approximate any continuous function arbitrarily well. Thus, we avoid  the difficulties arising from optimizing a large number of neural network parameters, as in SAC, while maintaining the expressive capacity necessary for effective shape modeling.

Thirdly, we propose a strategy to solve our regularized ERM problem in a RHKS  space  adapted to  the shape we would like to recover. This is possible thanks to the unique  correspondence between RKHS and kernels. Hence, instead of relying on handmade fixed kernels as in [76,78] , we learn the kernel parameters using a loss function that avoids overfitting on the data term (Equation 10 in the paper).

Regarding existing kernel reconstruction methods  we further note that:

- Both [76] and [78] address the problem of reconstruction from points and their normals. Differently, we use KRR to reconstruct from points only which is a much harder problem.
- Differently from [76] and [78], we propose a strategy to improve  generalization  by solving our regularized ERM problem (Equation 6) in a RHKS space adapted to the shape we would like to uncover This is possible thanks to the unique correspondence between RKHS and kernels. Hence, instead of relying on handmade fixed kernels as in [76] and [78], we learn the kernel parameters using a loss function that avoids overfitting on the data term (Equation 10). Additionally, we can dynamically adjust this function within the reproducing kernel Hilbert space by learning the Nystrom samples. This is a novel approach that was not explored in the context of shape reconstruction, as the resulting function from this optimization is more complex and expressive then a simple KRR mapping [76],78]. We will clarify further relation to [76] and [78] in our related work section.

### Why improve on an existing occupancy network ?

One might not have the important resources, time and data required to train a generalizable model from scratch. Alternatively, given readily available pretrained networks, we propose here an efficient and robust solution to improve their generalization at test time. We believe this strategy to be practical and useful for the community, and we hope this work will spur more contributions in the area of fine-tuning and transfer learning for shape reconstruction.

### Are improvements marginal ?

We outperform common state-of-the-art and improve considerably and consistently on our baseline on 3 datasets across all metrics. For the most important metric i.e. Chamfer distance (CD1), we offer an improvement of 22%in ShapeNet (Table 2), 15% in ScanNet (Table 5) and 19% in Faust (Table 6) wrt to our occupancy network baseline.

### Can we use other generalizable occupancy networks ?

Our method can be successfully applied to other existing occupancy networks. We show the result of combining the Shapenet trained network in Conv [55] with our strategy in Table 2 in the rebuttal PDF. We evaluate in the ScanNet benchmark as it is the most challenging in terms of generalization and our Conv [55] based method outperforms its baseline.

### Can we recover from the baseline’s failures ?

Figure 2 in the rebuttal PDF shows cases where we can obtain good reconstructions for baseline failures, which shows that our strategy of insuring inputs are at the levelset is effective.

### Additional Limitations

We provide additional limitations in the rebuttal PDF and we will discuss them further in the next version. First ablation of number of nystrom samples m in Table 3 shows that the KRR can possibly suffer from overfitting for large values of m, hence the choice of this hyperparameter can be challenging. Additionally, Figure 2 show an examples where, despite improving on the reconstruction of the network baseline, it is still challenging for our method to recover the finest structures and intricate details of complex shapes.

---

### Author Response · Authors · 2023-08-21
**Summary of changes**

We thank the reviewers for their positive reception of our work and appreciation of our rebuttal. Thanks to their valuable feedback from both the reviews and the discussion we improved the quality of our manuscript through the changes summarised below. More details on these elements can be found in the rebuttal.

**Introduction**: We clarified why generalizable shape networks can fail (underfitting the input). We clarified motivation of improving on pretrained shape networks (training such large models can be resource intensive, adaptation is less costly and yields considerable improvement), doing so with a learnable N-KRR (an alternative to unstable network weight tuning that offers a shape specific expressiveness/regularization tradeoff), and using a Gaussian kernel (universality properties of the associated RKHS, adaptation to preexisting features). We reorganized and improved writing.

**Related work**: We clarified relation and differences from [76,78] (although both our inputs and strategies differ, c.f. rebuttal), and introduced concurrent work NKRS CVPR23 (although our strategies differ). We added some missing weaknesses of prior work. We reorganized and improved writing.

**Method**: We made Fig. 1 more intuitive, and explained its color coding further. We reorganized and improved writing to be clearer.

**Results**: We added justification of choice of number of augmented points, comparison to SPSR, the generalizability result (using Conv [55] as a baseline for our method), ablation of number of inducing points “m” under optimization. We also added more visual results.

**Limitations**: We added a discussion of the effect of high “m” values on overfitting, a discussion and visual results of behavior under baseline failures, with the hypothesis and evidence of our improvement.

---

### Decision · Program_Chairs · 2023-09-21

**Decision:**

Accept (poster)

**Comment:**

This paper was reviewed by five knowledgeable referees. Initial feedback indicated that the approach is technically sound (JKo3, 7xQW), novel (Pm8Q, q2Vh), and boasts high accuracy (6hDU, q2Vh, LSLk). While the paper initially received varied scores, some reviewers expressed concerns regarding the motivation and justification for the technical choices (6hDU, 7xQW), as well as its comparison to some recent Neural kernel-based reconstruction works (LSLk, 7xQW). The author addressed these concerns, prompting all reviewers to revise and some to increase their scores, eventually reaching a unanimous decision for acceptance. After thorough consideration of the feedback, the AC agrees with the reviewers and has decided to accept this paper, with a strong recommendation for the authors to integrate the suggested changes into the final manuscript.